# Protective Immunity against Canine Distemper Virus in Dogs Induced by Intranasal Immunization with a Recombinant Probiotic Expressing the Viral H Protein

**DOI:** 10.3390/vaccines7040213

**Published:** 2019-12-10

**Authors:** Yanping Jiang, Shuo Jia, Dianzhong Zheng, Fengsai Li, Shengwen Wang, Li Wang, Xinyuan Qiao, Wen Cui, Lijie Tang, Yigang Xu, Xianzhu Xia, Yijing Li

**Affiliations:** 1Heilongjiang Key Laboratory for Animal Disease Control and Pharmaceutical Development, College of Veterinary Medicine, Northeast Agricultural University, Harbin 150030, China; jiangyanping@neau.edu.cn (Y.J.); jiashuo0508@163.com (S.J.); dianzhongzheng11@163.com (D.Z.); Yilvwenrou@126.com (F.L.); wangshengwen0518@163.com (S.W.); wanglicau@neau.edu.cn (L.W.); qiaoxinyuan@neau.edu.cn (X.Q.); cuien@neau.edu.cn (W.C.); tanglijie@neau.edu.cn (L.T.); yigangxu@neau.edu.cn (Y.X.); 2Institute of Military Veterinary, Academy of Military Medical Sciences, Changchun 130000, China

**Keywords:** canine distemper virus, H protein, probiotic vaccine, immunogenicity

## Abstract

Canine distemper virus (CDV) elicits a severe contagious disease in a broad range of hosts. CDV mortality rates are 50% in domestic dogs and 100% in ferrets. Its primary infection sites are respiratory and intestinal mucosa. This study aimed to develop an effective mucosal CDV vaccine using a non-antibiotic marked probiotic pPG^ΔCm^-T7g10-EGFP-H/*L. casei* 393 strain expressing the CDV H protein. Its immunogenicity in BALB/c mice was evaluated using intranasal and oral vaccinations, whereas in dogs the intranasal route was used for vaccination. Our results indicate that this probiotic vaccine can stimulate a high level of secretory immunoglobulin A (sIgA)-based mucosal and IgG-based humoral immune responses in mice. SIgA levels in the nasal lavage and lungs were significantly higher in intranasally vaccinated mice than those in orally vaccinated mice. Both antigen-specific IgG and sIgA antibodies were effectively elicited in dogs through the intranasal route and demonstrated superior immunogenicity. The immune protection efficacy of the probiotic vaccine was evaluated by challenging the immunized dogs with virulent CDV 42 days after primary immunization. Dogs of the pPG^ΔCm^-T7g10-EGFP-H/*L. casei* 393 group were completely protected against CDV. The proposed probiotic vaccine could be promising for protection against CDV infection in dogs.

## 1. Introduction

Canine distemper (CD) is an acute, highly contagious disease caused by the canine distemper virus (CDV) [1,2]. This disease spreads rapidly with a wide range of hosts and is distributed worldwide [3]. Infected animals can be characterized by symptoms such as fever, rash, diarrhea, nasal discharge, and conjunctivitis, and can also suffer occasional neurologic complications [4,5]. The mortality rate is closely dependent on the host species and can reach up to 50% in domestic dogs and 100% in ferrets [6]. CDV is transmitted by aerosols, and the virus enters the host through the nasal or oral routes. It initiates replication by contacting the signaling lymphocyte activation molecule (SLAM/CD150) receptor, which is expressed on the surface of immune cells, such as alveolar macrophages and/or dendritic cells of the respiratory tract [7,8]. Infected circulating immune cells disseminate the virus throughout the lymphatic system, followed by its spread to epithelial tissues by Nectin-4 [9]. The virus is amplified and secreted from the epithelial cells of the respiratory and gastrointestinal systems of the infected host [10]. In addition to respiratory and gastrointestinal symptoms, the disease is characterized by the rapid onset of severe leukopenia/lymphopenia and inhibition of lymphocyte proliferation during the initial weeks of infection [7,11]. Viral proteins are associated with modulation and inhibition of innate immunity within the host cells. In particular, the blocking of type I and II interferon (IFN) signal transduction is the main feature of CDV immunomodulatory activity [12,13].

At present, attenuated CDV vaccines that are widely used can significantly reduce the mortality rate in dogs; however, there are still reports of encephalitis induced by attenuated vaccines or CDV infection after immunization [14,15].Occasionally, protective immunity is not sufficiently induced when vaccines are administered in the presence of the maternal antibody, which is undetectable within 12 weeks [16,17]. In addition, a modified-live vaccine elicits robust immunity in dogs, but still retains sufficient virulence to cause severe disease and death in more susceptible species [2,18]. These disadvantages have led to an increased interest in developing a safe and efficient vaccine against CDV infection for highly susceptible species.

Lactic acid bacteria (LAB) is a genus of gram-positive facultative anaerobic or microaerophilic rod-shaped bacteria. These bacteria generally live in the digestive, respiratory, and genitourinary systems of humans and animals, without causing disease. Furthermore, they play an important role in the probiotic effects on the host. LAB can induce mucosal immune response and exert probiotic effects in both humans and animals [19,20,21]. This can be used to express functional proteins that carry pharmaceutical significance [22,23,24]. Moreover, this property makes them attractive candidates as antigen delivery carriers for the development of mucosal vaccines [25,26]. For mucosal immunization, LAB present a more attractive delivery system than other live vaccine vectors, such as *Shigella*, *Salmonella*, and *Listeria* [27,28], owing to the fact that they are considered safe, can be administered noninvasively (via oral or intranasal routes), and exhibit a mucosal adjuvant-like effect [20,21,29]. Moreover, specific *Lactobacillus* species were recently shown to specifically induce inflammatory responses against infection, increase immunoglobulin A (IgA) production, activate monocytic lineages [30,31], and regulate the balance between Th1and Th2 pathways [32,33].

Henceforth, considering the characteristics of CDV infection, developing a new vaccine that can induce specific secretory immunoglobulin A (sIgA) with neutralizing ability-based mucosal immune responses against CDV infection is of remarkable significance. In this study, a new approach to prevent CDV infection was explored using *Lactobacillus casei* 393, a potential antigen-delivery vehicle, to construct a genetically engineered pPG^ΔCm^-T7g10-EGFP-H/*L. casei* 393 strain expressing the H protein of CDV as a probiotic vaccine. Following intranasal immunization, the immunogenicity and immune protective effect of the probiotic vaccine were evaluated.

## 2. Materials and Methods

Animal experiments were performed in accordance with the international and national guidelines, OIE Terrestrial Animal Health Code CNAS-CL06:2018, respectively, for the care and use of laboratory animals. The protocol, 2017NEAU-124; 9 September 2017 was approved by the Committee on the Ethics of Animal Experiments of Northeast Agricultural University, Harbin, China.

### 2.1. Bacterial Strain, Virus, and Plasmid

*L. casei* ATCC 393 was cultured in de Man–Rogosa–Sharp (MRS) broth at 37 °C without shaking. CDV wild-type strain was obtained from primary canine kidney cells that had been cultured for seven generations after initial infection in a naturally infected domestic dog in 2016. CDV Snyder Hill strain kindly gifted by Professor Dongfang Shi, Northeast Agricultural University, was propagated on Vero cells (ATCC CCL-81) at 37 °C in a 5% CO_2_ incubator. The Vero were cultured in Dulbecco’s modified Eagle’s medium (DMEM) (Gibco, Carlsbad, CA, USA) which was supplemented with 10% fetal bovine serum (FBS) (Gibco). A constitutive expression plasmid, pPG-T7g10-PPT, was constructed at our laboratory [34]. This construct contained the gene which would express the PgsA anchor protein and also the T7g10 translation enhancer which would increase gene expression. The PgsA is a transmembrane protein derived from *Bacillus subtilis*, and the amino terminus of the protein of interest can be fused with PgsA and be firmly anchored on the surface of LAB [35]. The T7g10 effectively enhances translation and expression of specific proteins [36,37].

### 2.2. Construction of Recombinant Lactobacillus Strain

The genomic RNA from the wild-type CDV strain was extracted, followed by reverse transcription. The gene encoding the H protein of CDV was then amplified using PCR by using cDNA. The schematic diagram for the construction of the recombinant plasmid pPG^ΔCm^-T7g10-EGFP-H is shown in Figure 1. Briefly, the *H* gene (1824 bp), which would be transcribed and translated as it was within the entire open reading frame (ORF), was inserted into pPG-T7g10-PPT atSacI and ApaI sites to obtain the recombinant plasmid pPG-T7g10-H. Next, the gene encoding enhanced green fluorescent protein (EGFP) with a (GGGGS)3 flexible linker was inserted into the pPG-T7g10-H by using the SacI and KpnI sites, this would generate the plasmid, pPG-T7g10-EGFP-H. Subsequently, the chloramphenicol (*Cm*) resistance gene was deleted via enzyme digestion with NcoI and StuI, yielding the plasmid pPG^ΔCm^-T7g10-EGFP-H. Next, electroporation was performed as described previously [38], followed by a screening process using flow cytometry by using a FACSCalibur (BD Biosciences, San Jose, CA, USA) at 488 nm. The bacterial strain with the green fluorescence signal was collected and grown on an MRS plate at 37 °C for 36 h, yielding the recombinant strain, pPG^ΔCm^-T7g10-EGFP-H/*L. casei* 393, which was confirmed by PCR. All recombinant plasmids were identified by sequencing. Primers used in this study are listed in Table 1.

### 2.3. Identification of Proteins Expressed by the Recombinant Lactobacillus

The expression of proteins of interest was detected by growing the recombinant strain pPG^ΔCm^-T7g10-EGFP-H/*L. casei*393 within basal MRS broth at 37 °C for 16 h without shaking. The recombinant strain was harvested by centrifugation at 10,000× *g* for 2 min, followed by washing twice with phosphate buffered saline (PBS), and lysing with a Mini-Beadbeater (BioSpec, Bartlesville, OK, USA). After centrifugation, the supernatant was extracted and mixed with 5× sodium dodecyl sulfates (SDS) loading buffer and subsequently denatured in boiling water for 10 min. Then, they were analyzed using 10% SDS-polyacrylamide gel electrophoresis (SDS-PAGE). Next, proteins were transferred onto a polyvinylidene fluoride membrane, followed by immunoblot analysis by using mouse anti-EGFP monoclonal antibody (ZSGB-BIO, Beijing, China) or canine anti-CDV polyclonal antibody, prepared at our laboratory, as the primary antibody and HRP-conjugated goat anti-mouse/canine IgG antibody (Sigma, St. Louis, MO, USA) as the secondary antibody. The anti-CDV polyclonal antibody was derived from serum samples of the dog which was immunized with the inactivated CDV3 vaccine. The recombinant strain was prepared as described previously [39], and the expression of the EGFP protein on the surface of pPG^ΔCm^-T7g10-EGFP-H/*L. casei* 393 was observed using an ultra-high resolution microscope.

### 2.4. Immunization and Specimen Collection

The immunogenicity of pPG^ΔCm^-T7g10-EGFP-H/*L. casei* 393, via different vaccination routes, was evaluated by dividing a total of 240 6-week-old BALB/c mice into three groups. The mice were purchased from Liaoning Changsheng Biotechnology Co., Ltd., Liaoning, China. Each group was divided into two subgroups: The mice in one subset were immunized via the oral route and those in the other subset were immunized via the intranasal route. Each subset was further divided into three groups that received either, PBS, pPG-T7g10-PPT/*L. casei* 393 or pPG^ΔCm^-T7g10-EGFP-H/*L. casei* 393. Details of the immunization pathway and dosage are shown in Table 2. The immunization protocol was performed on days 1, 2, and 3. Booster immunization was administered on days 14, 15, and 16. A second booster was administered on days 28, 29, and 30 (Figure 2A). Serum and fecal samples were collected on days 0, 7, 14, 21, 28, 35, 42, and 49 after primary immunization. Samples were subsequently stored at −80 °C until use. Genital tract and nasal cavity secretions were obtained by washing the respective organs three times with 200 μL and 50 μL of sterile PBS, respectively [33], on days 0, 7, 14, 21, 28, 35, 42, and 49 after primary immunization, and stored at −80 °C until use. On days 0, 14, 21, 28, 35, 42, and 49 after primary immunization, five mice in each group were randomly killed to collect the sIgA sample from the lungs and intestine and stored at −80 °C until analysis. Of these, fecal samples used for detecting the levels of antigen-specific sIgA antibody were treated employing the method described previously with some modifications [22]. Briefly, 0.2 g of feces were suspended in 500 μL of PBS containing 1 mmol/L phenylmethylsulfonyl fluoride (PMSF) (Sigma, Ronkonkoma, NY, USA) and 1% bovine serum albumin. After the sample was incubated at 4 °C for 16 h and centrifugation at 12,000× *g* for 5 min, the supernatant thus obtained was stored at −20 °C until use.

The immunogenicity of pPG^ΔCm^-T7g10-EGFP-H/*L. casei* 393 in dogs was evaluated by randomly dividing a total of 32 female Beagle puppies into four groups. Beagle puppies were 5 weeks old, and serologically negative for CDV, canine parvovirus, and rabies virus. All Beagle puppies were purchased from Shifang Experiment Animal Corporation (Jiangshu, China). Two groups of dogs were immunized intranasally with either pPG^ΔCm^-T7g10-EGFP-H/*L. casei* 393 or pPG-T7g10-PPT/*L. casei* 393. Both groups received a dosage of 2 × 10^10^ CFU/kg. The immunization protocol of the probiotic vaccine was performed as described in Figure 2B. The dogs in one group immunized with PBS via the intranasal route were used as a negative control, and those in a group immunized with an attenuated CDV vaccine (CDV3) via intramuscular injection, which has been widely used in China, were used as a vaccine control. After primary immunization, serum, feces, eye secretion swab, nasal fluid swab, throat mucus swab, rectal mucus swab, and genital tract secretion swab were collected on day 42 (Figure 2B), and subsequently stored at −80 °C until use. After the swab samples were thawed, they were soaked in 0.5 mL Minimum Essential Medium (MEM) supplement. The samples were centrifuged, and the supernatant was placed in 96-well plates in triplicate (0.1 mL per well).

### 2.5. Enzyme-Linked Immunosorbent Assay (ELISA)

The levels of antigen-specific IgG and sIgA antibodies were determined using enzyme-linked immunosorbent assay (ELISA). Briefly, 96-well polystyrene microtiter plates were coated overnight with Snyder Hill strain of CDV propagated on Vero cells at 4 °C, followed by blocking with 5% skim milk at 37 °C for 2 h. Next, the collected samples as the primary antibody were added, and the plates were incubated at 37 °C for 1 h, followed by incubation with HRP-conjugated goat anti-mouse or canine IgA antibody (Sigma, USA) or IgG antibody (Sigma, USA) as the secondary antibody. Finally, the absorbance at OD_450*nm*_ was determined using *o*-phenylenediamine dihydrochloride (Sigma, USA) as a substrate. The levels of cytokine interleukin 2 (IL-2), IL-4, IL-10, IL-12, IL-17, and interferon γ (IFN-γ), in the serum of the samples collected on day 42 after primary immunization were, determined using an ELISA Kit (Biosource, San Diego, CA, USA).

### 2.6. Detection of Recombinant L. Casei 393 on Specific T Cell and Lymphocyte Proliferation in Mice

On day 40 after primary immunization, five mice from each group were killed for preparing splenocytes to detect specific T cell and lymphocyte proliferation. In brief, the splenocytes in quadruplicates at a concentration of 5 × 10^6^ cells/mL were cultured in a 96-well plate in RPMI-1640 containing 10% FBS at 37 °C with 5% CO_2_. For detecting specific T cells, 0.5 μL CD4-PE and CD8-APC antibodies to 500 μL was added to the cell suspension, and then incubated at 20 °C for 30 min. Cells were then washed three times with PBS (pH 7.2), and 200 μL cell suspensions (0.01 M PBS) were analyzed on a fluorescence-activated cell sorting (FACS) Calibur cytometer. Cell numbers were assessed using flow cytometry to ensure cell numbers higher than 10^4^ were present in every sample. For assessing lymphocyte proliferation the cells were stimulated with 1, 5, and 25 μg/mL of the H1 protein of DN2016 at 37 °C for 60 h, the H1 gene (1374bp), which would express the protein without a hydrophobic region at the N terminal of the H protein, was amplified by using recombinant plasmid pPG-T7g10-H as a template and H3/H4 as primers. The H1 protein was expressed as a soluble protein by pCold-H1/Rosetta in our laboratory. In parallel, re-stimulation with 5 μg/mL of concanavalin A and RPMI-1640 were used as positive and negative controls, respectively. With absorbance measured at OD_570*nm*_, lymphocyte proliferation was detected using the CellTiter 96 AQueous Non-Radioactive Cell Proliferation Assay (Madison, WI, USA), according to the manufacturer’s instructions. The stimulation index was calculated by dividing the mean reading of antigen stimulation wells by the mean reading of negative control wells.

### 2.7. Canine Distemper Virus (CDV) Neutralization by immune Dog Sera

CDV neutralizing antibody (NA) titers were determined in Vero cells using a TCID50 microtiter assay, as previously described [40], with minor modifications. Briefly, the 50% tissue culture infective dose (TCID50) values of Snyder Hill strain was detected using the Reed–Muench method. Serum antibodies collected from the vaccinated dogs on day 35 post-immunization were diluted at 1:10–1:320 (in a total of 100 μL) and subsequently incubated at 37 °C for 1 h. Next, the treated viruses were added to a monolayer of Vero cells cultured in 24-well plates and tested in quadruplicate. The cells were overlaid with 1% methylcellulose, and the plates were incubated at 37 °C in a 5% CO_2_ atmosphere. Cells were examined daily for 5 days in order to screen for CDV-specific cytopathic effects.

### 2.8. Challenge Experiment in Dogs

On day 42 after primary immunization, the puppies in each group were challenged with the CDV wild-type strain via the intranasal route. After the challenge, the dogs were monitored daily by measuring their body temperature and clinical examination until they were euthanized with Nembutal (Solabio Pharmaceutical Co. Ltd.,Beijing, China) at the end of the experiment. Small-volume blood samples were collected in Vacuette tubes (Greiner Bio-One, KremsmuÈnster, Austria) containing K_3_EDTA as an anticoagulant every second day after initial administration of the CDV wild type strain. Total lymphocyte counts were obtained using an automated counter (HEMAVET 950FS; Miami Lakes, FL, USA). The nasal swabs, eye swabs, throat swabs, rectal swabs, and blood were collected from all infected dogs every other day after infection until the end of experiment. On day 14 after the initial administration of the CDV wild type strain, the heart, liver, spleen, intestine, lungs, and bronchus of three puppies in each group were collected. These samples were stored at −80 °C until use. After the total RNA was extracted, the viral load in each tissue was detected using the quantitative real-time RT-PCR assay.

### 2.9. Real-Time RT-PCR (qRT-PCR) Analysis

Real-time PCR was performed in triplicate to determine the viral load in the heart, liver, spleen, intestine, lungs, and bronchus samples that had been collected, using the ABI Prism 7500 sequence detection system (Applied Biosystems, Foster City, CA, USA) with SYBR green fluorescence detection. Total RNA samples were extracted from the aforementioned tissues using TRIzol (Gibco) reagent according to the manufacturer’s instructions. Total RNA was reverse transcribed into complementary DNA (cDNA) using Moloney murine leukemia virus (M-MLV), reverse transcriptase, and Oligo(dT)_18_ Primers (Takara, Dalian, China). The cDNA was prepared for real-time qRT-PCR using a SYBR*^®^* qPCR Mix Reagent Kit (Takara). Real-time quantification PCR was performed to determine the absolute copy numbers of CDV in each tissue. A standard curve was generated by plotting the threshold values against the serially diluted plasmid DNA encoding the CDV H protein fragment. The primer pair, H5 and H6, was based on inserted gene H. A negative control reaction, lacking template DNA, was also as performed with using the above protocol.

### 2.10. Statistical Analysis

Data are shown as mean ± standard error of three replicates per test in a single experiment. Kaplan–Meier survival analysis was performed. Tukey’s multiple comparison tests were performed to analyze the differences between groups. *p* < 0.05 indicated significance, and *p* < 0.01 indicated high significance.

## 3. Results

### 3.1. Construction of the Recombinant Lactobacillus

Following the construction of recombinant plasmid pPG^ΔCm^-T7g10-EGFP-H and electroporation, the recombinant strain that displayed green fluorescence signal was screened using flow cytometry and cultured on an antibiotic-free MRS plate. The recombinant pPG^ΔCm^-T7g10-EGFP-H/*L. casei* 393 grew well on antibiotic-free MRS plates, but could not grow in the presence of Cm resistance, whereas pPG-T7g10-PPT/*L. casei* 393 and pPG-T7g10-EGFP-H/*L. casei* 393 grew well on the MRS plate irrespective of the presence or absence of Cm resistance (Figure 3A). Western blot results showed that the H protein of CDV and EGFP protein can be effectively expressed by the recombinant strain pPG^ΔCm^-T7g10-EGFP-H/*L. casei* 393 (Figure 3B). Moreover, direct observation using the Delta Vision OMX SR showed clear green fluorescence on the cell surface of pPG^ΔCm^-T7g10-EGFP-H/*L. casei* 393 (Figure 3C), but not on pPG-T7g10-PPT/*L. casei* 393.

### 3.2. Immune Responses Induced in Mice

To compare the levels of antibodies produced after both intranasal and oral route of immunization, this study first used mice as animal models. Specific antibodies were detected using ELISA. After a second booster immunization, antigen-specific mucosal sIgA antibody levels in the intestinal mucus (A, B), genital tract wash (C, D), nasal wash (E, F), lungs (G, H), and feces (I, J) of mice in the pPG^ΔCm^-T7g10-EGFP-H/*L. casei* 393 group were significantly higher than those of mice in the pPG-T7g10-PPT/*L. casei* 393 and PBS groups (*p* ˂ 0.01; Figure 4); antibody levels peaked at 35 days, indicating that the pPG^ΔCm^-T7g10-EGFP-H/*L. casei* 393 can effectively induce antigen-specific mucosal immune responses. In contrast, sIgA levels in the nasal fluid and lungs of mice immunized with the probiotic vaccine via the intranasal route were higher than those of the mice that received the probiotic vaccine via the oral route. However the sIgA level in the feces of mice in the oral route group was higher than that of the mice in the intranasal route group, and there was no significant difference in the levels of sIgA antibody between the genital tract and intestinal mucus of mice in either of the immunization route groups. Moreover, the highest levels of sIgA were noted in the feces when compared with those in the other tissues. The intestinal mucus for both immunization route groups and the nasal fluid for the intranasal group also produced high levels of IgA antibody. However, the high level was noted in the intestinal mucus compared with that in the other tissues. Moreover, antigen-specific IgG antibodies were detected on day 7 after primary immunization (Figure 4K,L). Following the booster immunization, the antibody levels in mice in both the intranasal and oral route groups increased and were significantly higher than those in the pPG-T7g10-PPT/*L. casei* 393 and PBS groups (*p* < 0.01), whereas no significant difference was observed between pPG-T7g10-PPT/*L. casei* 393 and PBS groups before and after immunization. This indicates that our probiotic vaccine can effectively induce antigen-specific humoral immune responses. Overall, our results showed that pPG^ΔCm^-T7g10-EGFP-H/*L. casei* 393 could induce mice to produce mucosal antibodies in the respiratory system after intranasal immunization.

### 3.3. Effect of Recombinant L. casei 393 on Specific T Cell and Lymphocyte Proliferation in Mice

On day 40 after primary immunization, we investigated whether pPG^ΔCm^-T7g10-EGFP-H/*L. casei* 393 can trigger a T cell response by detecting the concentrations of CD4^+^ and CD8^+^ T cells in spleen lymphocytes of mice by flow cytometry, and calculating the content ratio of the two. In the oral and intranasal administration groups, a higher ratio of CD4^+^/CD8^+^ T cells was noted in the spleens of mice vaccinated with pPG^ΔCm^-T7g10-EGFP-H/*L. casei* 393 than that in the control groups (Figure 5). Interestingly, a significantly higher frequency of CD4^+^/CD8^+^ T cells was observed in the intranasally vaccinated pPG^ΔCm^-T7g10-EGFP-H/*L. casei* 393 mice than that in the orally vaccinated mice (Figure 5). Moreover, the isolation of splenocytes from the mice in each group was stimulated with the purified H protein; significant splenocyte proliferation was detected in the pPG^ΔCm^-T7g10-EGFP-H/*L. casei* 393 group using the MTT assay, but not in the other two groups (Figure 6).

### 3.4. Cytokine Levels in Mice

On day 40 after primary immunization, the levels of IFN-γ, IL-2, IL-4, IL-10, IL-12, and IL-17 in the sera of mice in each group were determined; compared with those in the pPG-T7g10-PPT/*L. casei* 393 and PBS groups. Significant increases (*p* < 0.01) in IFN-γ, IL-2, IL-4, IL-10, IL-12, and IL-17 levels were noted in the pPG^ΔCm^-T7g10-EGFP-H/*L. casei* 393 group, indicating that the recombinant strain could induce Th1, Th2, and Th17 cell immunity (Figure 7). According to the ratio of IL-4/IFN-γ, the Th2 immune response showed predominance. In addition, the concentrations of these cytokines in the sera of mice in the pPG-T7g10-PPT/*L. casei* 393 group were notably higher than those in the PBS group, suggesting that the nonengineered probiotic itself can directly enhance the nonspecific immunity of the body.

### 3.5. Immune Response Induced in Dogs.

The immune response induced by intranasal immunization of dogs was evaluated by collecting serum, feces, rectal mucus, genital tract mucus, eye secretions, nasal fluid, and throat mucus samples of the puppies in each group on day 42 after primary immunization. This was followed by detection of antigen-specific IgA and IgG antibody levels in these tissues by using ELISA. A significant level (*p* < 0.01) of IgG antibody was induced in pPG^ΔCm^-T7g10-EGFP-H/*L. casei* 393 group and the vaccine control group compared with that in the pPG-T7g10/*L. casei* 393 and PBS groups (Figure 8A). Moreover, the levels of IgA antibody in rectal mucus, feces, genital tract, eye secretions, nasal lavage, and throat mucus of puppies in the probiotic vaccine group and also in the commercial vaccine control group were significantly higher (*p* < 0.01) than the PBS groups; in addition, there was high levels of sIgA in dogs administered with pPG-T7g10-PPT/*L. casei* 393, which were equivalent to those induced by CDV3 in rectal mucus, eye secretion, feces, and throat mucus. The levels of the IgA antibody in the rectal mucus and nasal lavage were higher than those in the other tissues. Moreover, sera collected were used for CDV NA assays. Our study showed that dogs vaccinated with pPG^ΔCm^-T7g10-EGFP-H/*L. casei* 393 and CDV3 had higher levels of NA of CDV in their serum (Figure 8B), and their levels were significantly different from those in the pPG-T7g10-PPT/*L. casei* 393 and PBS groups (*p* < 0.05).

### 3.6. Protection Afforded by the Probiotic Vaccine against CDV Challengein Dogs

The protection efficacy of the probiotic vaccine against CDV infection in dogs was evaluated by challenging the dogs in each group with CDV on day 42 after the primary immunization via the intranasal route and observing the clinical symptoms of all animals daily. The results indicated (Figure 9) that all dogs in the commercial vaccine group and the pPG^ΔCm^-T7g10-EGFP-H/*L. casei* 393 group showed no significant clinical symptoms after challenge with CDV, whereas dogs in the pPG-T7g10-PPT/*L. casei* 393 and PBS groups developed symptoms of canine distemper, including high body temperature, lymphopenia, viremia, and respiratory tract problems. Body temperature curves showed that dogs in the pPG-T7g10-PPT/*L. casei* 393 and PBS groups developed fever with a biphasic thermal response, in accordance with the feature of canine distemper (Figure 9A), whereas no obvious signs were noted in any dogs of the pPG^ΔCm^-T7g10-EGFP-H/*L. casei* 393 and vaccine groups. Lymphocyte counts of blood (Figure 9D) were strongly reduced on day 4 after infection in the pPG-T7g10-PPT/*L. casei* 393 and PBS groups and did not recover during the second week, whereas the number of lymphocytes increased briefly on day 2 after infection in the pPG^ΔCm^-T7g10-EGFP-H/*L. casei* 393 and vaccine groups, and then returned to normal. Systemic and local virus replication in each group was detected using qPCR. Virus levels in blood markedly increased in the PBS and mock-vehicle groups on day 4 after challenge; peak viremia levels were observed around 6 ± 8 days (Figure 9C). CDV RNA was not detectable in the blood of the vaccine group, however a narrow-range increase was noted in the pPG^ΔCm^-T7g10-EGFP-H/*L. casei* 393 group; virus was detected from eye swabs, nose, throat, and rectal in the pPG-T7g10-PPT/*L. casei* 393 and PBS groups (Figure 9B). Viral RNA was detected in these samples as early as on day 3 after challenge; in most dogs, progressively increasing viral loads were noted during 7–8 days after challenge. Whereas CDV RNA was not detectable in the vaccine group, However, a minor increase was noted in the pPG^ΔCm^-T7g10-EGFP-H/*L. casei* 393 group. Moreover, viral loads in the heart, liver, spleen, intestine, lungs, and bronchi of puppies in each group were detected using qPCR; viral load in pPG^ΔCm^-T7g10-EGFP-H/*L. casei* 393 group and the commercial vaccine control was significantly lower than that of puppies in the PBS and pPG-T7g10-PPT/*L. casei* 393 groups (Figure 9E). Cumulative mortality of dogs in each group is shown in Table 3.

## 4. Discussion

CDV is an enveloped negative-strand RNA virus, which belongs to *Morbillivirus* genus of the Paramyxoviridae family along with the measles virus [41]. H is a protein that is located on the surface of the CDV particles and mediates attachment to the host cell via a receptor [5,41,42,43]. The H protein has many viral-neutralizing epitopes that can induce the production of neutralizing and protective antibodies as it plays an important role in humoral and cellular immunity [1,17,42]. A CDV F and H protein-expressing canarypox-based vaccine has been licensed for several years now and has been successfully used in various wildlife species [44,45]. Thus, H is an important target for vaccine development. *Lactobacillus* is widely used as a live vaccine vehicle against various microbes [23,46]. Appropriately, we constructed a recombinant plasmid with the EGFP screening marker, pPG^ΔCm^-T7g10-EGFP-H, which was transferred into *L. casei* 393 plasmid, which is an important probiotic bacteria and commonly used in fermented dairy products and functional foods [47]. Previous studies suggest that *L. casei* 393 possesses immunomodulatory, anti-inflammatory, and anti-tumor properties [48,49].

Mucosal vaccines cause low stress in animals and have been the subject of growing interest owing to the advantages that they offer over conventional vaccines. Another advantage is the stimulation of mucosal immune responses [50,51]. Mucosal vaccines are administered mainly orally or intranasally, and the route of immunization has a primary role in the induction of protective immune responses against challenge by pathogenic microorganisms. In this study, for evaluating the antibody levels induced by recombinant bacteria through different immune pathways, first, BALB/c mice were used as an animal model for vaccination with pPG^ΔCm^-T7g10-EGFP-H/*L. casei* 393 via the intranasal and oral routes. We found that the probiotic vaccine could effectively induce an IgG-based humoral immune response and IgA-based mucosal immune response. This was seen in both the oral and intranasal administration routes used for vaccination. Moreover, induction of sIgA levels in the nasal lavage and lungs of mice was higher in the intranasal group than in the oral immunization group, whereas sIgA antibody levels in the feces of mice in the oral group were higher than those in the intranasal group. Intestinal mucus was also found to produce higher levels of IgA antibody regardless of vaccination route. As expected, antibody titer levels were higher at the site of inoculation than at the remote site. No significant difference was observed in the levels of sIgA antibody in the intestinal mucus of mice when comparing the two immunization route groups. This finding may, in part, be due to the fact that the vaccine could have entered the digestive system through the esophagus during nasal immunization. A antibody levels in feces was the highest compared to the other tissues of mice that were sampled. It appears that there are non-specific reactions that occur in feces due to the complexity of feces composition.

Cell-mediated immune responses also played a crucial role in protecting the host from invading pathogens. As a result, we analyzed lymphocyte proliferation, T-helper cell immune responses, and cytokine activation after immunization. Lymphocyte proliferation positively correlated with cellular immunity and our findings also showed that the H protein significantly promoted splenic lymphocyte proliferative reactions, which suggested that a cellular immune response can be initiated as a result of our probiotic vaccine. Furthermore, CD4+ and CD8+T cells are important effector cells for cell-mediated immunity. In the present study we show that immunization with recombinant lactobacillus promoted the proliferation of CD4+ and CD8+ T lymphocytes. The levels of cytokines indicated the recombinant LAB could induce Th1, Th2, and Th17 cell immunity (Figure 7). According to the ratio of IL-4/IFN-γ, the Th2 immune response showed predominance. Th1 cells are important in macrophage activation and cell mediated defense against intracellular pathogens, which include viruses, bacteria, parasites, and yeast. Th1 cells are also implicated in graft rejection and some forms of autoimmunity [52]. In contrast, Th2 cells promote the production of most immunoglobulins, mediate defense against extracellular parasites, and are involved in a number of allergic responses. Cytokines produced by Th1 cells can negatively regulate the function of Th2 cells, while cytokines that are produced by Th2 cells can regulate the function of Th1 cells [53,54]. Moreover, Th17 is believed to play an important role in the protection against extracellular pathogens and development of autoimmune disorders [53,55,56,57]. In addition, the concentrations of these cytokines in the sera of mice in the pPG-T7g10-PPT/*L. casei* 393 group in the present study were notably lower than those in the pPG^ΔCm^-T7g10-EGFP-H/*L. casei* 393 group. These findings suggest that the H protein itself can directly enhance the cell immunity. Previous studies in our laboratory have also demonstrated this phenomenon [23,24,25].

In mice that were intranasally vaccinated, we found that our probiotic vaccine could induce both antigen-specific IgA-based nasal and intestinal mucosa immune response. Hence, the CDV antibody-negative weaned puppies were used to evaluate the immunogenicity of the pPG^ΔCm^-T7g10-EGFP-H/*L. casei* 393 via intranasal vaccination. In parallel, a commercial CD vaccine was used as a vaccine control by injection. The result indicated that the probiotic vaccine could effectively induce antigen-specific humoral and mucosal immune responses in dogs via intranasal immunization. Subsequently, the NA titer was detected; its value in all dogs of the pPG^ΔCm^-T7g10-EGFP-H/*L. casei* 393 group was relatively higher than that in the PBS and pPG-T7g10-PPT/*L. casei* 393 groups. In addition, significantly higher levels of IgA were noted in dogs administered with pPG-T7g10-PPT/*L. casei* 393 (Figure 8A), which in fact were equivalent to those induced by CDV3 in rectal mucus, eye secretion, feces, and throat mucus, Thus, pPG-T7g10-PPT/*L. casei* 393 probably induced dogs to produce non-specific IgA. LAB can interact with antigen-presenting cells (APCs) such as dendritic cells (DC) and induce sIgA and IgG. Hence, the mechanism of DC activation and the resulting immune responses are highly dependent on the LAB strain [20,29].

To evaluate the protective efficacy of the pPG^ΔCm^-T7g10-EGFP-H/*L. casei* 393 as a mucosal vaccine against CDV infection, we established an infection model on day 42 after primary immunization in dogs with CDV wild-type strain. We found that dogs in the pPG^ΔCm^-T7g10-EGFP-H/*L. casei* 393 group with NA titers above 40 could obtain the same effective immune protection as in the vaccine group; no signs of viremia or systemic infection were found. The results of virus loads in different tissues indicated that dogs vaccinated with pPG^ΔCm^-T7g10-EGFP-H/*L. casei* 393 were protected against CDV infection as well. The dogs immunized with PBS and pPG-T7g10-PPT/*L. casei* 393 developed persistent lymphopenia and typical clinical symptoms for CDV infection, and they produced large amounts of viral RNA (Figure 9B–E). These results indicated that nasal pPG^ΔCm^-T7g10-EGFP-H/*L. casei 393* vaccination relieved the disease symptoms, and that systemic immunity was stimulated by specific mucosal immunity. The final mortality rate was 100% in the PBS and pPG-T7g10-PPT/*L. casei* 393 groups, and that of the vaccine and probiotic groups was 0. Noticeably, CDV RNA was not detectable in the blood, eye, nose, throat, or rectal swabs obtained from the vaccine group. However, a minor increase was observed in the probiotics group (Figure 9B,C). This could be due to the fact that the dogs were indeed infected during the challenge with CDV; however, the virus was quickly cleared by the body.

Most pathogens enter the body at mucosal sites and protection of these barrier tissues is mediated by innate and adaptive immune responses. Antigen-specific antibodies and cell-mediated responses are the main defenses of the adaptive response. Therefore, the most efficient way to induce innate and adaptive mucosal immune response is to immunize directly on the mucosa rather than through systemic routes such as parenteral injection [56,57]. In this study, the recombinant probiotic vaccine with good immunogenicity could effectively induce antigen-specific IgA-based mucosal and IgG-based humoral immune responses and subsequently increased protection against the pathogen that we had administered. Moreover, after analysis of sIgA levels in the nasal cavity and lungs, it was found that these areas are resistant to invasion of CDV. Compared with traditional vaccines, LAB vaccines can also induce serum antibodies and systemic cell-mediated responses. Mucosal delivery is a particularly attractive mechanism of vaccination as it is relatively easy to administer. Furthermore, the common-mucosal immune system that is activated allows for the initiation of an immune response that occurs at one mucosal surface to be followed to immune cells of other distant mucosal sites [56]. Additionally, it could also cause the body to produce cell-mediated immune responses. Nonetheless, before releasing the probiotic vaccine for commercial clinical, further investigations are required.

## 5. Conclusions

In summary, a genetically engineered probiotic vaccine expressing the H protein of CDV was constructed in this study. The immunogenicity in mice or dogs that were intranasally or orally vaccinated with the probiotic vaccine was evaluated. The immune protection for dogs against CDV challenge was also evaluated. Our results reveal superior immunogenicity for the recombinant probiotic vaccine and prove that can effectively induce antigen-specific IgA-based mucosal and IgG-based humoral immune responses. Additionally, it can provide immune protection against CDV infection in dogs, substantiating is role as a promising mucosal vaccine.

## Figures and Tables

**Figure 1 vaccines-07-00213-f001:**
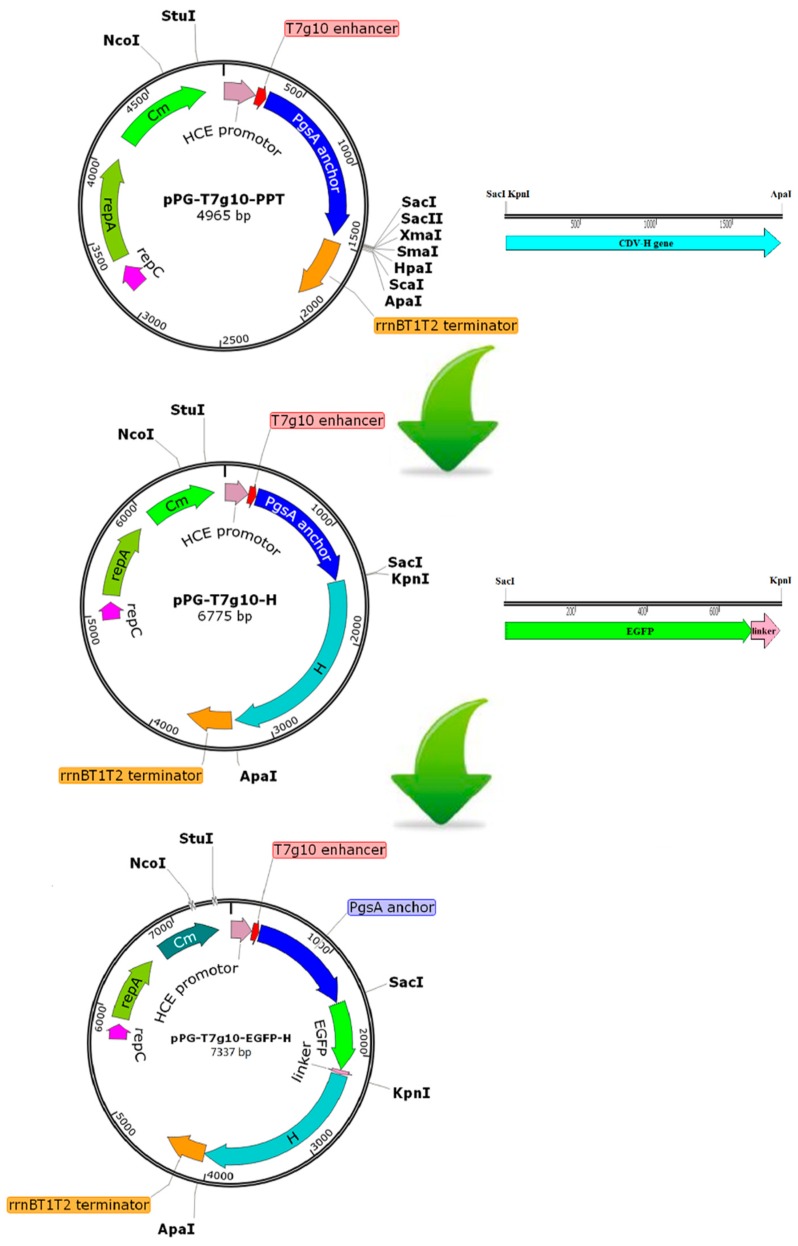
Construction of recombinant expression plasmids in this study.

**Figure 2 vaccines-07-00213-f002:**
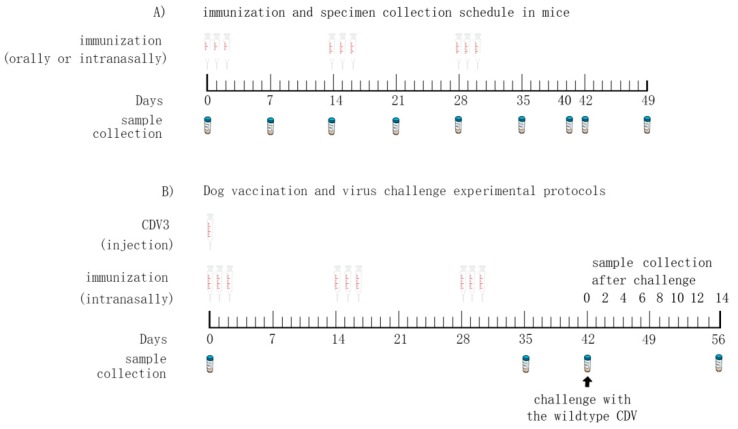
Immunization protocol and specimen collection for mice (**A**) and puppies (**B**).

**Figure 3 vaccines-07-00213-f003:**
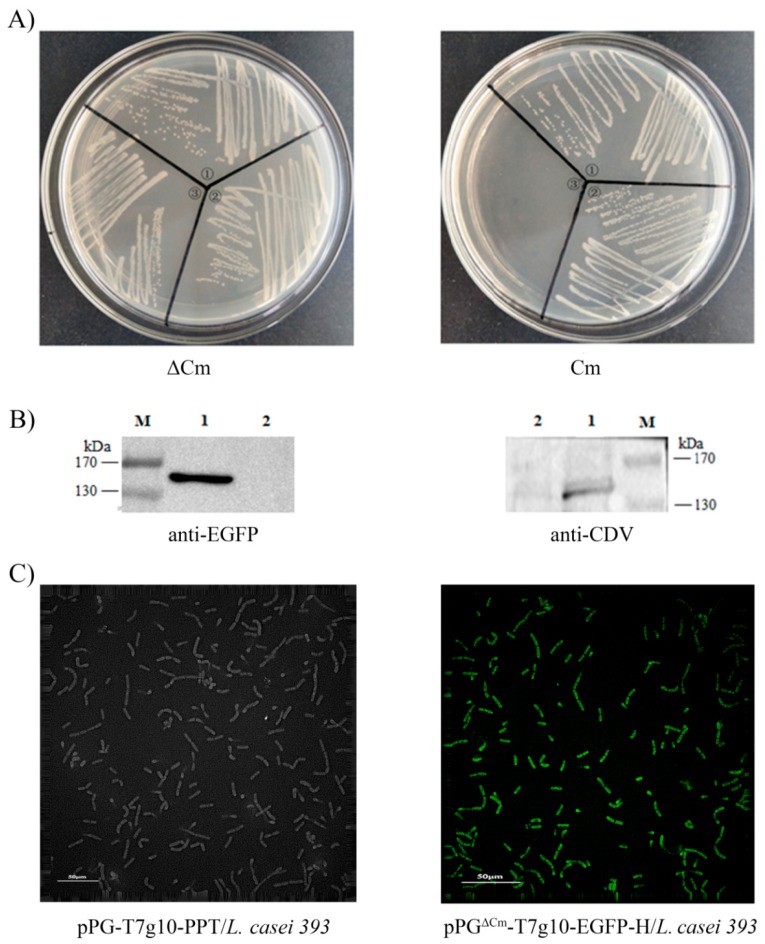
Of the recombinant strain pPG^ΔCm^-T7g10-EGFP-H/*L. casei* 393 and identification of protein expression. (**A**) Identification of chloramphenicol deletion in the recombinant pPG^ΔCm^-T7g10-EGFP-H/*L. casei* 393. ① pPG-T7g10-PPT/*L. casei* 393; ② pPG-T7g10-EGFP-H/*L. casei* 393; ③ pPG^ΔCm^-T7g10-EGFP-H/*L. casei* 393. (**B**) Identification of protein expression by using western blot. M: Protein marker; 1: pPG^ΔCm^-T7g10-EGFP-H/*L. casei* 393; 2: pPG-T7g10/*L. casei* 393. (**C**) EGFP expression observed using an ultrahigh resolution microscope.

**Figure 4 vaccines-07-00213-f004:**
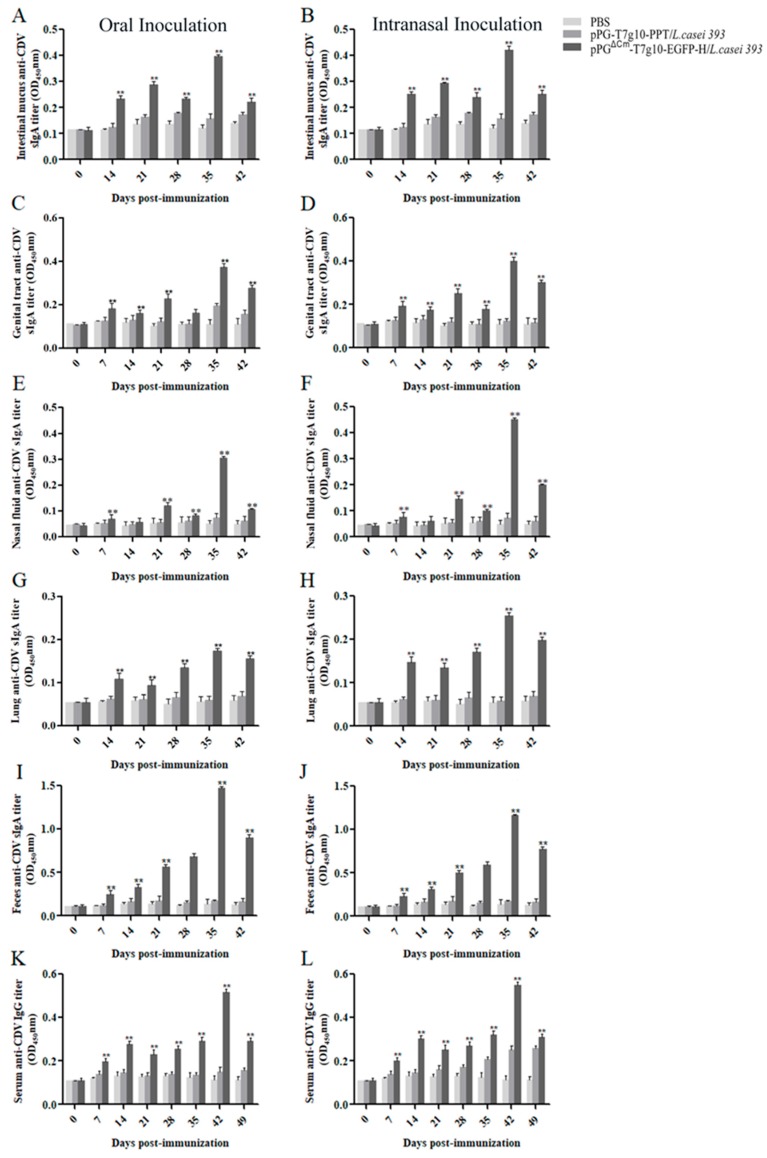
Levels in BALB/c mice immunized with the probiotic vaccine via the intranasal and oral route. After primary immunization, samples were collected on days 0, 7, 14, 21, 28, 35, and 42, followed by the detection of antigen-specifics IgA levels in the intestinal mucus (**A**,**B**), genital tract wash (**C**,**D**), nasal wash (**E**,**F**), lungs (**G**,**H**), and feces (**I**,**J**) and those of antigen-specific IgG in the serum (**K**,**L**) by using enzyme-linked immunosorbent assay (ELISA). Bars represent the mean ± standard error in each group (* *p* < 0.05, ** *p* > 0.01 as compared with pPG-PPT/*L. casei* strain 393 and PBS groups).

**Figure 5 vaccines-07-00213-f005:**
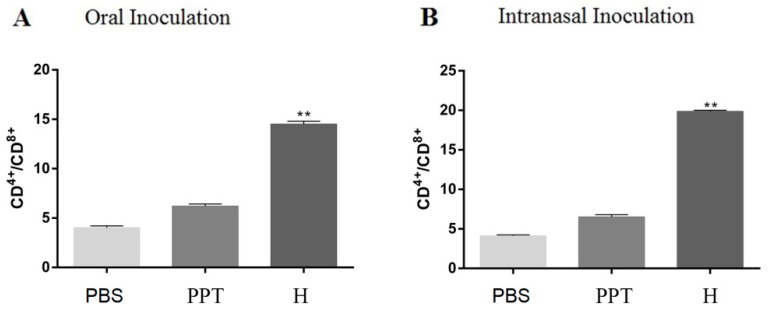
CD4^+^/CD8^+^ in splenic lymphocytes. Mice were immunized orally (A) or intranasally (B). Detected the concentrations of CD4+and CD8+ T cells in spleen lymphocytes of mice by flow cytometry on day 40 after primary immunization.(PPT) pPG-T7g10-PPT/*L. casei*393, (H) pPG^ΔCm^-T7g10-EGFP-H/*L. casei*393. (** *p* < 0.01 compared to the control groups: pPG-T7g10-PPT/*L. casei*393 and PBS).

**Figure 6 vaccines-07-00213-f006:**
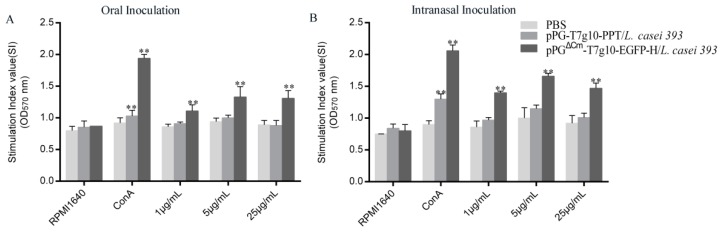
Lymphoproliferation within the mice in each group was detected using MTT assay by using CDV H protein as a stimulating agent. Mice were immunized orally (**A**) or intranasally (**B**). Bars represent the mean ± standard error value of each group (* *p* < 0.05, ** *p* < 0.01 compared to the control groups: pPG-T7g10-PPT/*L. casei* 393 and RPMI1640).

**Figure 7 vaccines-07-00213-f007:**
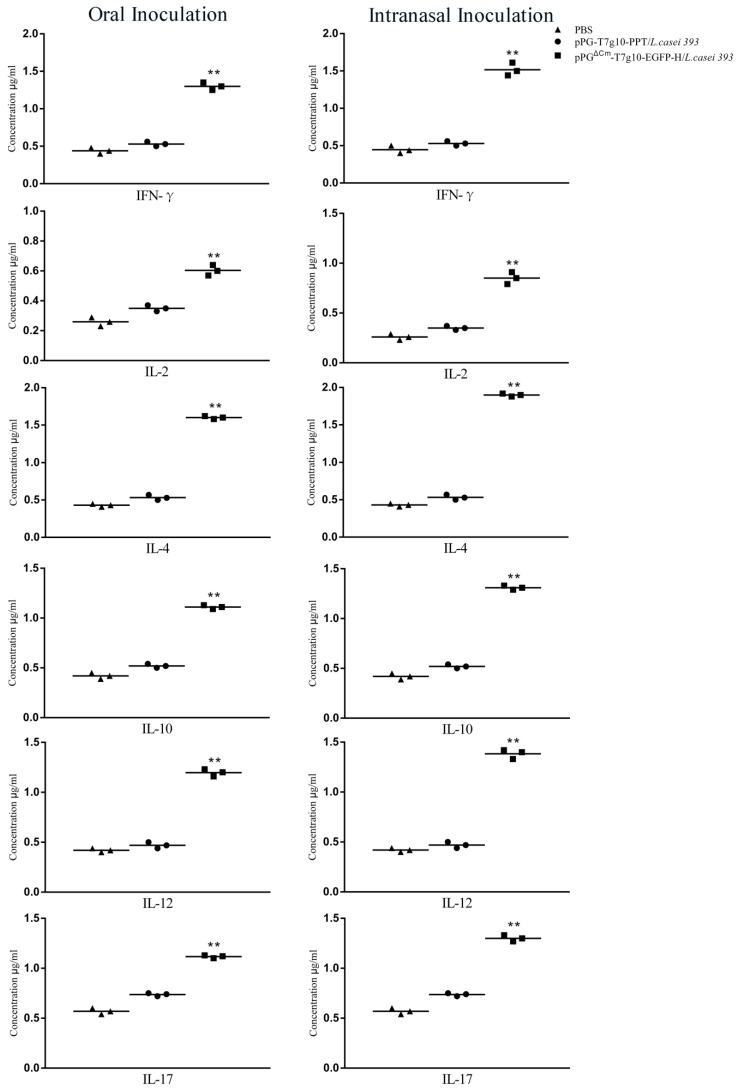
The levels of cytokines in the serum of immunized mice. On day 42 after primary immunization, the serum samples were collected from the mice in each group, and the concentrations of cytokines IFN-γ, IL-2, IL-4, IL-10, IL-12, and IL-17 were determined using an ELISA kit. Results are mean ± standard error in each group (* *p* < 0.05, ** *p* > 0.01 as compared with pPG-T7g10-PPT/*L. casei* 393 and PBS groups).

**Figure 8 vaccines-07-00213-f008:**
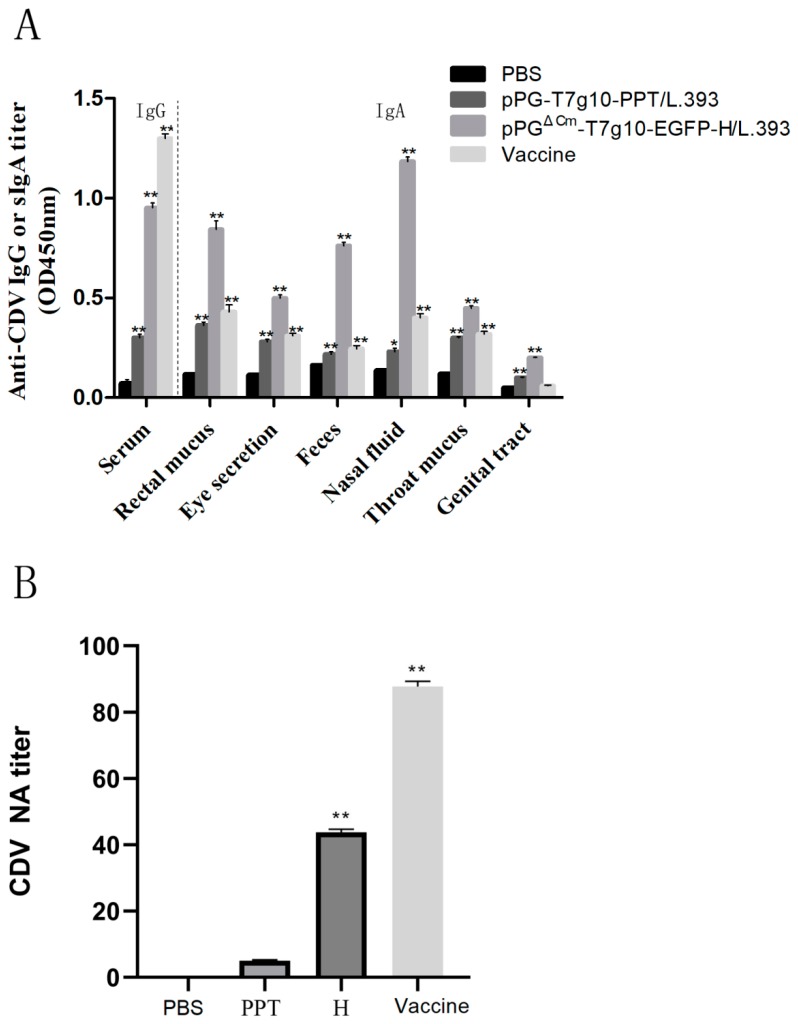
Immune response induced in dogs. The levels of IgG and IgA antibodies in dogs in each group on day 42 after immunization detected using ELISA (**A**). Blood samples were collected to detect neutralizing antibodies (NA) to CDV (**B**) at 35 days after vaccination. (L.393) *L. casei*393, (PPT) pPG-T7g10-PPT/*L. casei*393, (H) pPG^ΔCm^-T7g10-EGFP-H/*L. casei*393.Bars represent the mean ± standard error value of each group (* *p* < 0.05, ** *p* < 0.01 compared to the control groups: PBS and pPG-T7g10-PPT/*L. casei*393).

**Figure 9 vaccines-07-00213-f009:**
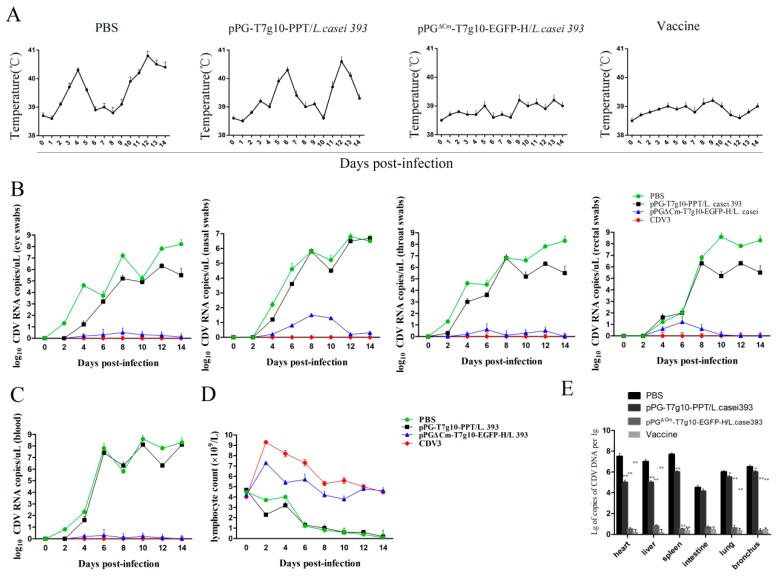
Protection effect of the probiotic vaccine against CDV in dogs. All dogs in the commercial vaccine and the pPG^ΔCm^-T7g10-EGFP-H/*L. casei* 393 groups showed no significant signs after challenge with CDV, whereas dogs in the pPG-T7g10-PPT/*L. casei* 393 and PBS groups developed symptoms of canine distemper, including high body temperature (**A**), local virus replication (**B**), viremia (**C**), and lymphopenia (**D**). Viral loads in different tissues were determined using qPCR (**E**). * *p* < 0.01, ** *p* < 0.001.

**Table 1 vaccines-07-00213-t001:** Primers used in this study.

Target Gene	Primers	Sequences (5′–3′)	Application
H	H1	GAGGAGCTC^1^CTCGGTACCATGCTCTCCTACCAAGACAAG	For lactic acid bacteria expression
	H2	CTTGGGCCC^1^TCAAGGTTTTGAACGGTTACATGAG	
	H3	CCAGGATCC^1^ATGGTTGGGGTCAAAAAATCTAT	For *E. coli* expression
	H4	CTTGTCGAC^1^TCAAGGTTTTGAACGGTTACATGAG	
	H5	TGACAGCAACGGTTCACAAGATGG	For qRT-PCR
	H6	CAGAGACCAATACAGGCACCATCC	
EGFP	E1	ATGGTGAGCAAGGG	For amplification of EGFP
	E2	TCACTTGTACAGCTCGTC	

^1^ Restriction enzyme recognition sites used for cloning are underlined.

**Table 2 vaccines-07-00213-t002:** pathway and dosages for mice.

Group Pathway		Dose
PBS	Oral	200 μL
	Intranasal	20 μL
pPG-T7g10-PPT/*L. casei* 393	Oral	10^10^ CFU/200 μL
	Intranasal	10^10^ CFU/20 μL
pPGΔCm-T7g10-EGFP-H/*L. casei* 393	Oral	10^10^ CFU/200 μL
	Intranasal	10^10^ CFU/20 μL

**Table 3 vaccines-07-00213-t003:** Protection efficiency against CDV challenge in dogs intranasally immunized with recombinant pPG^ΔCm^-T7g10-EGFP-H/*L. casei* 393.

Group	Immune Route	No.	Deaths	Survival	Cumulative Mortality
PBS	Intranasal	8	8	0	100%
pPG-T7g10-PPT/*L. casei* 393	Intranasal	8	8	0	100%
pPGΔCm-T7g10-EGFP-H/L. casei 393	Intranasal	8	0	8	0%
Vaccine	Intranasal	8	0	8	0%

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
