# Peer review of "Protective Immunity against Canine Distemper Virus in Dogs Induced by Intranasal Immunization with a Recombinant Probiotic Expressing the Viral H Protein"

_vaccines, 2019, doi:10.3390/vaccines7040213_

Round 1

Reviewer 1 Report

Jiang et al. reported a probiotic vaccine for canine distemper virus (CDV) infection using Lactic acid bacteria. In general, the study has performed well and systematically using well-established methods. The probiotic vaccine successfully induced the CDV-specific mucosal immunity in mice and dogs, and the data clearly demonstrated the sufficient efficacy of the probiotic vaccine against CDV infection in dogs. Therefore, this concept of vaccine can be possibly used for the future vaccine for CDV in dogs. However, there was certain inconsistency between the presented data and authors’ description in the manuscript. Also, certain important information was missing in the current form of the manuscript.

Major comments

Please indicate the detailed information of the H gene region, which was inserted in the plasmid. Was it the entire open reading frame of the H protein, only ectodomain, or a partial H gene region? The strain name and genotype information of the CDV strain for the H protein is useful for specific readers. Please provide with the detailed information of the anti-CDV polyclonal antibody, which was prepared in the authors’ laboratory (line 128). Was the same immunization protocol used for dogs as that was used for mice? (line 159) Please provide with the strain information of CDV used for making ELISA plate. The strain name and genotype information of the CDV strain for the H protein is useful for specific readers. Please provide with the detailed information of the H protein produced by using E. coli; methods, which were used, the CDV strain name (genotype) for the H protein etc. (line 194) Which CDV strain (name, genotype) was used for the neutralizing assay? (line 204) Which CDV strain (name, genotype) was used as the challenging virus? (line 213) The detailed information of the quantitative real-time RT-PCR is necessary. The information of ‘Oral’ and ‘Intranasal’ is missing in Figure 4. The description about the data in Figure 4 (line 259–263) is seemingly and partially inconsistent with Figure 4. Was the sIgA level in the feces of mice in the oral route group higher than that of the mice in the intranasal route group? Was the level indeed highest in the intestinal mucus? The level in nasal fluid was also high. High levels of cytokines were detected only in the EGFP-H/L group, but not in PPT/L group (lines 306–314). Did these data suggest that the H protein itself stimulated the cytokine responses? If so, what mechanisms can be propsed? Figure 8A showed significantly high levels of CDV-specific IgG and sIgA in dogs administered with PPT/L393. The levels of antibodies were equivalent to those induced by Vaccine (CDV3) in rectal mucus, eye secretion, feces, and throat mucus. Were they correct? Why did a low level, but significant level, of neutralizing antibody against CDV was detected in dogs with PPT/L? (Figure 8B) Please carefully check the consistency between the description in lines 409–415 and data in Figure 4. The levels of viral load were different between probiotic vaccine and CDV3 vaccine groups (line 446–447). Viral RNA was undetectable in CDV3 vaccine groups, whereas certain levels of viral RNA, albeit very low, were detected in probiotic vaccine group (Figure 9).

Minor comments

‘Special’ antibodies? It may be ‘Specific’. (line 253) Viruses were ‘detected’ by RT-PCR, but not ‘isolated’ (line 366). Viral RNA was detected, but not infectious virus (line 449). The H protein mediates attachment to the host cell by binding to a receptor. The membrane fusion process is not required for the attachment. (line 388)

Reviewer 2 Report

Jiang et al. present a novel CDV vaccine based on a recombinant probiotic L. casei strain which can be administered intranasally or orally.
The idea is to increase mucosal immunity against CDV H.

CDV is a relevant veterinary pathogen which infects which enters the host at mucosal interfaces, thus such vaccines are of high interest.

Jiang et al. test their probiotic vaccine in mice and dogs by intranasal and oral routes.
They show that both vaccination routes induce sIgA at different sites and IgG in Balb/C mice.
Vaccination leads to increased CD4/CD8 ratios in spleen and increased proliferation of splenocytes stimulated with CDV H.
Further, elevated cytokine levels are reported after immunization of mice compared to control mice treated with PBS or L.casei with empty plasmid.

Jiang et al. then compare their novel probiotic vaccine to a commercial CDV vaccine in dogs, showing lower serum IgG levels, but higher sIgA levels
at different sites after vaccination with the recombinant L. casei CDV H vaccine.
Vaccinated dogs were then challenged with CDV. While control dogs succumed to disease, both the commercial and the novel vaccine were protective.

Overall this is a coherent manuscript. However, I have several questions and suggestions:

In the Materials and Methods section some more details are required:
- Line 93 origin of CDV sample
- Line 95 details of CDV propagation
- Table 1: which of these primers were used for which experiments?
- Lines 128-129: details on antibodies used
- Line 194: recombinant protein production
- Lines 223-224: details of qRT-PCR

The term "high significance" for p values below 0.01 does not make sense.

Some of the Figures should be amended:
- Figure 4: indicate which diagrams refer to intranasal and oral vaccineation (as done in Figure 7)
- Figures 4-7: Was there only one PBS control group that is displayed in both the diagrams for intranasal and oral vaccination? This should be stated.
- Figures 4,5,7: Y-axes of left and right diagrams should have the same hight and intervals to facilitate comparison
-

Several points require additional clarification and discussion:
- Line 73: Provide a brief explanation why LAB is more attractive
- Could the addition of EGFP in the plasmid influence immunogenicity?
- Figure 7: The vaccine seems to induce Th1, Th2 and Th17 cytokines. The implications of this finding including possible advantages and disadvantages should be discussed.
- Figure 8: Also L. casei electroporated with empty plasmid seems to increase anti-CDV antibodies. This requires clarification/discussion.
- Line 365: "one dog had been infected" - what does this mean?

- Lines 392-393, use of LAB as vaccine vehicle: please provide some references/examples
- Lines 395-398: explanation of PgsA and T7g10 could be moved to the introduction or the methods section where the plasmid is presented

- What is the authors' overall conclusion on utility of the vaccine and advantages/disadvantages over the commercial vaccine? This discussion of these points should be expanded
with reference to the data presented in the manuscript.

Minor points:
- Abstract line 32 "virulent 42 days": there is a word missing
- Line 253: should probably read "specific", not "special" antibodies
- Line 285 "contents" does not seem to be the appropriate term
- Line 408-409 "both the immune pathways" - does this refer to both routes of vaccination?

Round 2

Reviewer 1 Report

The authors have revised almost sufficiently, according to the reviewers' comments.

Howver, this reviewer still has a question about the data in Figure 8. The authors discussed about the sIgA induced by PPT/L.393 in the discussion section of the revised manuscript. Was it 'specific' or 'non-specific antibody?

The below is the sentences by the authors.

In addition, there was significantly high levels of 'CDC-specific sIgA' in dogs administered with PPT/L 393.

This seems that PPT/L.393 induced dogs to produce 'non-specific IgA'.

Author Response

Dear reviewer,

    We would like to thank you for giving us constructive suggestions which would help us to improve the quality of the paper. We have revised the paper according to your comments. In addition, the English language of the current manuscript has been polished and improved. We have marked all the changes in the revised manuscript. We appreciate your warm work earnestly, and hope that the correction will meet with approval.

We have revised the manuscript accordingly, and detailed corrections are listed below point by point:

Point 1: This reviewer still has a question about the data in Figure 8. The authors discussed about the sIgA induced by PPT/L.393 in the discussion section of the revised manuscript. Was it 'specific' or 'non-specific antibody? The below is the sentences by the authors. In addition, there was significantly high levels of 'CDC-specific sIgA' in dogs administered with PPT/L 393. This seems that

Response 1: Thank you for your careful work. Yes, PPT/L.393 induced dogs to produce 'non-specific IgA. We have deleted the related content in result and discussion (Lines 364 and Lines483).